# Hidden stories of caregivers with children living with sickle cell disease in Uganda: Experiences, coping strategies and outcomes

**Isaac Alinda**[1]*, **Lydia Kabiri**[1], **Hadad Ssebagala**[2]

**1** Department of Nursing, School of Health Sciences, College of Health Sciences, Makerere University, Kampala, Uganda, **2** School of Medicine, College of Health Sciences, Makerere University, Kampala, Uganda

* alindaisaac53@gmail.com

## Abstract

### Background

Sickle Cell Disease (SCD) poses a substantial public health challenge in Uganda, exhibiting distinct regional variations in prevalence. The Uganda Sickle Cell Surveillance Study has estimated an overall SCD prevalence of 13.3%. Notably, this prevalence varies significantly across the country's regions, reaching its highest in the northern region at 22.2% and it's lowest in the central region at 5.3%. This variation reflects the diverse impact of SCD and underscores the need for a comprehensive understanding of its regional implications. SCD places substantial physical, social, and psychological burdens on caregivers, potentially leading to heightened parental stress. However, limited research has focused on the daily challenges and experiences of SCD caregivers, despite evidence showing the detrimental impact on their emotional well-being, personal lives, employability, and socioeconomic status. This study explored the undisclosed struggles of Ugandan caregivers tending to children with SCD, uncovering their coping strategies and subsequent outcomes.

### Methods

In June and July 2023, we conducted in-depth interviews with caregivers at Mulago sickle cell clinic after obtaining their informed consent. An interview guide was used as the primary data collection tool, with interviews lasting 30–45 minutes. Twelve participants were recruited, ensuring comprehensive data collection by following the data saturation principle. We analyzed the collected data using open coding.

### Results

Three key themes emerged: caregiver experiences, coping strategies, and outcomes. Initially, caregivers grappled with confusion and uncertainty before a formal diagnosis. Financial strain and inadequate support posed persistent challenges, affecting their emotional well-being. Coping strategies varied, encompassing traditional remedies and

**Data availability statement:** All relevant data are within the paper and its Supporting Information files.

**Funding:** financial support granted by the Fogarty International center of the national institutes of health, US, department of state's office of the U.S global AIDS co-coordinator and health diplomacy (S/GAC) and the president's emergency plan for AIDS relief (PEPFAR), HEPI SHSSU (Health professions education and training for strengthening the health system and services in Uganda) under grant No. 1R25TW011213 The funders played an active role in supporting the study by providing financial resources for data collection, stationery procurement, participant facilitation, data analysis, and dissemination of the study findings. However, they had no role in the study design, data analysis and interpretation, decision to publish, or preparation of the manuscript.

**Competing interests:** The authors declare that they have no competing interests.

modern medical treatments for symptom relief. Coping outcomes were complex, reflecting caregivers' resilience alongside substantial emotional distress and sleep disturbances. The substantial financial burden further exacerbated their overall well-being.

## Conclusion

Elevating awareness and knowledge about sickle cell disease within communities is essential. Such awareness can empower caregivers of children living with sickle cell disease, promoting emotional resilience and mitigating family disruptions.

## Introduction

Sickle cell disease (SCD) is one of a set of inherited red blood cell abnormalities characterized by the substitution of hydrophobic valine for hydrophilic glutamic acid in position 6 of the beta-globin chain. With about 400,000 newborns affected by sickle cell disease each year and roughly 300,000 of these having sickle cell anemia, sickle cell disease is a significant worldwide health concern [1].

Most sickle cell diseases occur in sub-Saharan Africa, where over 75% occur; this percentage is expected to rise by 2050 [2]. The prevalence of sickle cell disease in Uganda being the fifth highest in Africa means that more effort needs to be done to promote sickle cell awareness in Uganda communities as per the emphasis on the inclusion of sickle cell in health education campaigns [3]. Children with sickle cell trait comprise 13.3% of the Ugandan population, while those with sickle cell disease constitute 0.7% among the cohort of samples tested in 2015 [1] another study conducted in 2019 with the majority of the children tested between 5–9 months, of which the cohort of samples referred specifically for sickle cell testing, the overall prevalence of sickle cell disease was 9.7% and particularly elevated in high-burden districts where focused screening occurred [4]. In addition, according to rough estimates of birth rates, Uganda will see 16 695 newborns with sickle cell illness and 236 905 newborns with sickle cell trait per year. Although they had varied topographical distributions, alpha-thalassemia trait and Glucose-6-Phosphate Dehydrogenase Deficiency (G6PD) were frequently observed alongside sickle cell illness in children aged 0–24 months [4]. In order to effectively manage sickle cell disease in a resource-constrained environment, parenting a child with the condition in Uganda requires a comprehensive strategy that addresses the disease's logistical, emotional, and physical issues. Taking care of a child with sickle cell disease can be expensive, and many families in Uganda may find it difficult to pay for the required therapies and drugs. In addition, the financial burden of sickle cell disease treatment varies significantly by region. The East Central and Mid-Northern regions carry the highest burden of SCD, with the estimated costs per detected case ranging from $278.07 USD in East Central to as much as $2607.19 USD in the South Western region. The average cost per detected SCD case across Uganda is approximately $483.74 USD. This variation reflects how regions with lower SCD prevalence incur higher costs per positive case detected, compared to regions with a higher disease burden [5].

One study looked at the various coping techniques used by caregivers of children and adolescents with SCD 0 to 19 years old and the psycho-social issues they face in Eastern Uganda, this research revealed that the psycho-social difficulties faced by caregivers of children and teenagers with SCD in Uganda were complex and included emotional, social, and economic difficulties. In addition, that the caregivers employed a variety of coping strategies to overcome these difficulties, acceptance, others still lived in denial, while the rest used talking with others and getting counseled to reduce the intensity of the feelings experienced [6].

The stress of financially and emotionally supporting children with SCD impacts caregivers' quality of life, which is likely to impact their level of social and professional success [7]. There is evidence that caregiver health issues may affect the way they care sickle cell patients. Frequent hospital visits and disturbances in caregiver lifestyle, relationships, and interests were predictors of emotional distress among caregivers, which affected four out of every ten of the participants examined [8].

Caregivers employed a combination of adaptive and maladaptive coping techniques, which may include religious coping. They linked the application of adaptive coping to improved patient-related results and reduced psychological distress in caregivers. Conversely, a higher frequency of maladaptive coping was associated with unfavorable outcomes for both patients and caregivers [9]. The Ministry of Health in Uganda has developed policies, standards, screening programs, and preventive measures to enhance the treatment and management of sickle cell disease. However, there remains a significant gap in the continuum of care for sickle cell patients in Uganda, particularly concerning the provision of psychosocial support services for caregivers. To address these challenges effectively, there is a pressing need for increased awareness and education among both healthcare professionals and the general population.

Getting a deeper understanding of the factors that impact the quality of life for caregivers of children with sickle cell disease is paramount. This understanding will enable the development of strategies that are more supportive and effective, ultimately improving the well-being of both caregivers and patients. The rationale for conducting this study stems from the absence of published research in Uganda focusing on the experiences of caregivers responsible for children with sickle cell disease (SCD). Managing SCD in Uganda presents unique challenges, which may lead caregivers to harbor 'hidden' experiences that require exploration. This study aims to shed light on their coping strategies and the outcomes of these strategies, filling a critical knowledge gap in the country.

## Methods

### Study design

Our research employed a qualitative approach with a phenomenological orientation to delve into the untold narratives of caregivers, explaining their coping strategies and the resulting outcomes of caring for children living with Sickle Cell Disease at the Mulago Sickle Cell Unit in Kampala, Uganda. The research adhered to the guidelines outlined in the Consolidated Criteria for Reporting Qualitative Research (COREQ) with specific consideration of three key facets: 1) the composition and introspection of the research team; 2) the methodological framework employed; and 3) the comprehensive presentation of research outcomes [10].

### Study setting

The investigation took place within the confines of the Sickle Cell Unit at the Mulago National Referral Hospital. This medical facility is on Mulago Hill, found in the northern sector of Kampala, approximately 5 kilometers northeast of the city's central business district. The Sickle Cell Clinic at this hospital operates during weekdays, providing both inpatient and outpatient services to patients and their caregivers.

It is important to note the context of sickle cell disease (SCD) management in Uganda. Taking care of a child with SCD can be financially challenging for many families in Uganda due to the high costs associated with required therapies and drugs. This economic burden often poses significant challenges to accessing timely and appropriate care for affected children.

## Participant eligibility and recruitment

We employed purposive sampling to select 12 participants, comprising 2 males and 10 females, with ages ranging from 27 to 52 years. Our selection focused on caregivers who had children diagnosed with sickle cell disease and had engaged in caring for the patient as part of their regular care routine. It was essential that the chosen caregivers were not providing care to bedridden children during data collection. Importantly, no participant declined to participate in our study, and all caregivers approached consented willingly to take part. Among the 12 participants, 5 caregivers had children aged below 6 years, 2 handled children aged between 7 and 12, while the remaining 5 caregivers cared for children aged 13 to 19. Our selection aimed to ensure the representation of caregivers spanning all age groups, acknowledging that experiences can differ based on the age of the children under their care.

The sample size for this study was determined based on the principle of data saturation, which is a commonly applied concept in qualitative research to determine the adequacy of the sample size. Data saturation refers to the point at which no new information or themes are observed in the data, indicating that the sample size is sufficient to capture the range and depth of experiences related to the research question [1,11].

To ensure data saturation, we employed an iterative process of data collection and analysis. After each interview, the data were transcribed and analyzed to identify emerging themes and patterns. As the study progressed, we continuously compared new data with previously collected data to determine if new insights were still being generated. This iterative process allowed us to monitor data saturation and make informed decisions regarding the adequacy of our sample size.

Additionally, we conducted member checking with participants to validate our findings and ensure that we had captured the full range of experiences related to the caregiving role for children with sickle cell disease. This process further supported the achievement of data saturation, as participants confirmed that no new themes or insights were emerging from the data [12].

## Qualitative data collection

**Interview guide development.** We constructed a semi-structured interview guide through a synthesis of insights gained from previous participant interactions, an extensive review of relevant literature, and the application of the transactional model of stress and coping. To ensure effective communication with both participants and researchers in Kampala, we made the interview guide available in both English and Luganda languages. Prior to the main data collection, we conducted a pretest of the interview guide at the Mulago sickle cell clinic, involving two participants from the study. This pretesting phase allowed us to fine-tune the questions, ensuring their relevance and comprehensibility. Feedback from this pilot phase was meticulously considered, resulting in adjustments or removal of specific questions from the guide.

**Data collection.** Between June and July 2023, the principal investigator, Alinda Isaac (A.I) conducted in-depth interviews, with local assistance from research associates. COVID-19 safety measures were diligently observed throughout the data collection, encompassing social distancing and the consistent use of face masks. The interviews took place at a designated clinic meeting area, chosen for its spaciousness, privacy, and freedom from potential disturbances such as noise.

Participants willingly provided both oral and written consent prior to their involvement in the study, with research associates and the principal investigator collecting the written consents. To protect participant anonymity, no personal identifying information was recorded or transcribed. We duly secured permission for audio recording the individual interviews. All interviews occurred within the premises of the Mulago sickle cell clinic and spanned between 30 to 45 minutes. To further safeguard participant anonymity, each participant was assigned a

pseudonym. However, in the results section, participant numbering was used (such as, Participant 1, Participant 2) to maintain consistency with the raw data and ensure confidentiality.

We stored data securely to ensure confidentiality and privacy.

**Data analysis.** The audio records of the interviews were meticulously transcribed into Luganda and subsequently translated into English. The transcription and translation processes were conducted by qualified translators under the supervision of the principal investigator, A.I., to ensure accuracy. Additionally, participants were consulted during the translation to maintain the authenticity and meaning of their responses.

The analysis was conducted using a reflexive thematic analysis approach, which emphasizes the importance of reflexivity in the analysis process. This approach involves continuously reflecting on our own perspectives, biases, and assumptions throughout the analysis to ensure a rigorous and transparent approach. The analysis comprised the following stages, adapted from Braun and Clarke's method as per their 2019 publications on reflexive thematic analysis: becoming acquainted with the data; generating initial codes; identifying relevant themes; and reviewing and refining these themes [13].

We subjected the transcripts to a comprehensive dual reading and manual coding process, overseen by the principal investigator, A.I., and the supervisor, Lydia Kabiri, (L.K). Through open coding, codes were established, allowing for the exploration of new ideas and concepts. While the initial coding centered on the language and expressions used by the participants, these codes facilitated the identification of prominent concerns and the development of a theoretical framework. We methodically arranged and categorized codes generated according to their similarities, ultimately forming core themes.

Concurrently, relevant literature concerning caregivers' experiences, coping strategies, and coping outcomes was consulted throughout the data analysis. This external knowledge aided in the comprehension and refinement of the emergent themes. We employed a conceptual mapping process to explain the relationships and associations between these themes, aligning with the study's focus.

## Ethical considerations

This research received ethical approval from the Uganda National Council of Science and Technology and Makerere University School of Health Sciences Research Ethics Committee under the reference number MAKSHSREC-2023-509. We got administrative clearance from the Mulago National Referral Hospital Ethics and Research Committee, as well as the authorization of the Sickle Cell Unit in-charge.

We diligently secured written informed consent from the caregivers, emphasizing their absolute freedom to withdraw their participation at any point during the study. Importantly, their decision to participate or not had no bearing on the healthcare provided to the adolescents in the hospital. To ensure privacy, participants' identities were protected by replacing their names with initials, and any other personal information about caregivers was appropriately concealed.

Participants were reassured that ample time would be allocated during the interviews to alleviate stress, encourage the open expression of their concerns, and ease any hesitations. In creating a safe and supportive environment, the researchers exhibited empathy and afforded participants their full and undivided attention throughout the interviews.

## Results

The results of the interviews including illustrative quotes of the participants are presented. The participants included chefs, business women, peasants, school teachers, farmers, self-employed and Stay-at-home parents: The hidden stories of caregivers of children living with

sickle cell disease; with a focus on exploring their experiences, identifying their coping strategies and examining the resulting outcomes are discussed. Three themes were identified: 1) experiences; 2) coping strategies; and 3) coping outcomes.

## Experiences

Financial challenges were a recurring theme among the caregivers. Many struggled to afford proper medical care and treatment for their children with limited financial resources. Some participants mentioned receiving occasional support from family and friends, but the overall financial burden remained significant.

**Financial stress and support: (N = 12).** nearly all participants underscored the substantial financial burden linked to the care of children with sickle cell disease. They recounted stories of resorting to borrowing money from friends and resorting to land sales to meet the escalating medical bills and transportation expenses. Participants further elaborated on the hardships related to accessing healthcare services, especially for various age groups and the severely ill, because of the remote locations that caused arduous journeys on foot to reach the nearest healthcare facility.

> **Participant 8 :** *"Finances... disturbs us a lot, ever since I saw it... I have ever sold my land to spend over 4 million to reach him where he is..."*

> **Participant 9;** *"Sometimes the money that you could have for him to take him to school you take it to the hospital. Sometimes we reach here while I admit you and there is no medicine.."*

> **Participant 11**; *"Even when the husband has no some money, you can be worried, deep inside your heart, while you say that where is the man going to get the man, while I am going to take the child to the hospital, and I even buy the medicine.."*

Most of the participants lamented the burden of the wholesome money spent on the diet and nutrition on the affected children since the times of diagnosis, since extra care and precaution has to be taken because of their delicate nature, any time they can get illness, or an attack,

> *"…you have to make sure that atleast he eats anything that he asks from you, since you delay, things may not end well,… I have to make sure that he eats a balanced diet on every meal…"* participant 4

**Uncertainty and confusion prior to diagnosis.** Several participants recounted the initial perplexity they experienced concerning their baby's symptoms. Family beliefs about the incessant cries of the baby sometimes reached such levels that it resulted in marital strains and, sometimes, led to divorces. The futile attempts to seek remedies from nearby clinics, which proved unsuccessful, eventually led most caregivers to the referral hospital. It was only they made there that a definitive diagnosis and treatment begun. Notably, a lack of awareness about the condition was prevalent among the parents of most participants during this period.

> **Participant 1** expressed, *"For me, I had to first witness it on this last born, when I gave birth to him, he had 5.30 kg, when he reached the 6th month, he reduced in size, swelling of the fingers, toes and then we brought him here at Mulago and they came to find out that he had sickle cell."*

> *".. the beginning, we didn't know what was the exact cause of his illness while every time he used to be sick, now I think like in 2022 in November, that's when we came to know of it…"*
> **Participant 2**

*".. That's how it can be, he sick for most of the times, he is absent from school, every time you with him at home, while he is appetite-less, he cannot drink, you are the one who spends the entire night seated and looking, whereas for the man, pushes it to your side saying that you are the one who birthed such children who look like that.."* **participant 4**

However, most of the participants shared about the use of the modern medicines got from the hospital premises after the diagnosis, despite the influence on the administration and trial of the traditional medicines of treatment.

*".. the 11th month, that's when he got the attack, but when he got the attack, we came here and was bedridden, and we stayed here, the health care workers taught us what to do, for me that's what I have been following, and he hasn't yet got the attack,"* **participant 3**

**Social stigma.** In terms of their neighborhoods and social interactions, many participants expressed feelings of inadequacy, particularly when surrounded by healthy children. These sentiments extended beyond the hospital setting to their homes, where the unpredictable SCD attacks of their children sometimes led them to conceal the illness to ward off the perceived "evil eye." Some caregivers also shared experiences of insensitive behavior from their neighbors, which left them disheartened.

*"There is a time when a neighbor came at home and said, 'that child, pay my mother, but for you, you are a young child, why are you despising me?' Then he replied; 'now how have I undermined you?' Then the neighbor replied, 'for you, you may not even reach my age.' So that child had to burst out and cry and said, 'but why are you telling me such?"* **Participant 5.**

However, the half of the participants (n = 6), elaborated how their neighbors influence was quite neutral as far as the child's illness is concerned and others lamented how they didn't pay attention to whatever they did about their family having a child with sickle cell.

## Coping strategies

**Coping strategies at home (n = 11).** caregivers employed various strategies at home to manage their child's illness. These included administering painkillers, traditional herbal remedies, and nutritional support to ease symptoms and enhance their child's well-being.

Participant 10 uses various herbal and local remedies, such as hibiscus, beetroot, and black jack leaves, to manage their child's sickle cell symptoms. They try to use these natural remedies to ease pain and improve their child's health when medical treatments are unavailable or limited.

*".. most of the times we cook some herbal medicines, sometimes I get those pain killers as they are the ones I have been walking with, this is because I have not been having some money to take them there and sometimes the one to bring me to Mulago,…"* **Participant 1**

*".., I give him some avocado, I cook it, black jack leaves like that, and give him, because sometimes there is when he gets sick and he cries and complains of dizziness and you see he is weak, and yet you don't have the Ugx 10,000 to come to the hospital, then I say to myself, let me first give him some "ekisepiti", but obviously, I see that it (herbal remedies), have yielded no major outcome.."* **Participant 5**

**Emotional coping and encouragement played a crucial role.** Since it was the only ray of light for most of the participants, living within, working towards the best of the family and better health outcome for both the caregiver and the children in the family.

Participant 5 expressed, *"I am aware of the body-building food, I am aware of the drinks they can eat so that they have better health, that's how I have been moving in such a situation while am working with such strategies, to prevent the pain…"*

**Coping strategies at school.** Participants encountered challenges in ensuring their child's education while managing the disease. They employed various strategies, such as pleading with school authorities and teachers for understanding and flexibility.

**Participant 1**; *".. for even for the school fees for the previous term, I was overwhelmed, and they didn't give them their report cards, however, they went there and studied but for the report card, they didn't get it, and even for this term too, I have not paid it, and even this one too, he hasn't gone for the studies too, whereas I don't have any strategies that I have put in place…."*

*"…. then you can go talk to the school nurse, you explain to her the way, she will give her the medication, the way you also treat her at home, whereas for such a nurse, you cannot just tell her all those words like someone passing laws at home, and yet you as a person, cannot get some money to reward her…"* **Participant 4**

**Social support (n = 9).** the level of social support varied among caregivers. While some received emotional encouragement and help from their support networks, others faced stigmatization and negative reactions from their communities amidst the care for the child.

*"…. there is one time when my neighbor, over abused my son, because there was a nearby goat that was sick, he started calling my son that sick goat, such a thing hurt me so bad, but I said to myself, in these, God is the all-knowing,…."* **Participant 5**

*"…the good thing that is there is that, the neighbors know it he is sick, and I even have a neighbor who normally brings for me, the pawpaw leaves, those which are dry because I once told her we needed them a lot…"* **participant 3**

*"… I didn't know that the people had known it he was sick, I was there while, I just saw them coming, those to give him money gave it, and even the teachers came to check on him, and gave him some support…"* **participant 3**

*"…. for me, for the friends' they have been there for me because whenever they see my son is sick, they stand with me, even when I'm penniless, sometimes they give me money and then I come to the hospital, sometimes amongst his relatives, o. k just to say that the child will not heal, I rarely disturb them in that whenever I come to the hospital…"* **Participant 5**

## Coping outcomes

**Emotional coping and acceptance (n = 12).** All participants expressed emotional distress and anxiety related to their child's condition. They described the initial shock and confusion upon learning of the sickle cell diagnosis. Over time, many caregivers showed acceptance with making hard decisions and resilience in managing their emotions.

**Participant 4,** "…. *things that helped me a lot after they had told me and I got to know that he is having sickle cell disease, the first thing I did was, I asked the Almighty to forgive me, since I decided it for myself, that I will never produce again, and by then I was pregnant preparing for the second baby, and then I said to myself that I would not produce a 2ⁿᵈ child because for the 1ˢᵗ child made me spend most of the times in the hospital while bed ridden.…*" The participant shares their emotional journey of acceptance after learning about their child's sickle cell disease. In addition, the participants express seeking forgiveness, deciding about family planning, and finding forgiveness for others who may not fully understand the challenges they face.

Participant 5, "…*I can observe it he is not growing big... I can encourage him... even if I check my pocket and I see I am not having money, I just say 'God, the child is sick, however, I don't know how I am going to do it'…*"

**Participant 7**; "…*Because you can go through a lot...you are crying because of the situation that you can go through...for me, I really don't see it where a person says that he could be happy, because yourself in your heart you don't like yourself, now like for me, I don't like myself…*"

**Hope for the child's future.** most of the participants shared about the living hope, nothing else to be done since they are not the ones that brought the disease, it's God's plan since, and we only hope for the best so that they can also grow up and becomes individuals of importance for the future generations. Participant 9 expressed hope for their child's future despite the challenges posed by sickle cell disease.

".. *I want him to go back [to school], because he is too worried, he says that the whole year, and yet in the future I will have like 2 years off…*" This theme reflects the caregivers' determination to provide the best possible life for their children.

However, some caregivers experienced positive outcomes such as improved health, reduced frequency of illness;
Participant 4 reflects on the positive outcomes of their coping strategies, including improved health, reduced frequency of illness, and the ability to maintain a semblance of normalcy in their child's life despite the challenges. "… *I have yielded good results because even while she is at school, most of them are unaware because they don't see her falling sick, even the ones that I meet here they tell me well for her she doesn't resemble the rest..*"

## Discussion of results

### Interpretation of findings

The analysis of data from caregivers of children living with sickle cell disease in Uganda revealed a complex and challenging landscape for these caregivers. The themes that emerged shed light on their experiences, coping strategies, and coping outcomes, providing valuable insights into the realities they face.

### Social demographic data

The socio-demographic data of the 12 participants in this study provide valuable insights into the characteristics of caregivers of children living with sickle cell disease in Uganda. Interestingly, there were only 2 male participants as shown in Table 1, highlighting the gender dynamics that shape caregiving roles within households. Traditionally, women have taken on

the majority of newborn care tasks, while men often fulfill the role of economic providers and decision-makers regarding family health [14]. This gendered division of labor and responsibilities may explain the underrepresentation of male caregivers in this study. However, future studies need to find the other perceptions of male population to participate in the direct care for these children living with sickle cell to increase the male representation and improvement of both maternal and child health in Uganda.

Moreover, the varied years of duration of care reported by the participants, with the highest being around 20 years as shown in Table 1, demonstrate a high level of commitment towards ensuring better health outcomes for children living with sickle cell disease. This finding suggests that caregivers in Uganda have been persistently managing the challenges posed by the disease for extended periods, emphasizing the long-term nature of caregiving.

## Experiences

In comparing the findings with existing research on caregivers of children with chronic illnesses or sickle cell disease in other regions, several similarities and differences emerge. Like previous studies, caregivers in Uganda face financial challenges and emotional strain in caring for their children with sickle cell disease. The reliance on traditional remedies aligns with the

**Table 1. Participants demographic characteristics.**

| Baseline characteristic | Frequency (%) | No. of participants |
|---|---|---|
| **Sex** | | |
| Female | 83.3 | 10 |
| Male | 16.7 | 2 |
| **Age of the caregiver (years)** | | |
| 25–34 | 25 | 3 |
| 35–44 | 58.3 | |
| 45–54 | 16.7 | 2 |
| **Marital status of the caregiver** | | |
| Married | 75.0 | 9 |
| Divorced | 16.7 | 2 |
| Widow | 8.3 | 1 |
| **Duration of care** | | |
| < 10 | 25.0 | 3 |
| > 10 | 75.0 | 9 |
| **Employment** | | |
| Unemployed | 25.0 | 3 |
| Trading | 25.0 | 3 |
| Farmer | 16.7 | 2 |
| Teacher | 8.3 | 1 |
| Peasant | 25.0 | 3 |
| **No. of children with sickle cell** | | |
| 1 | 91.7 | 11 |
| > 1 | 8.3 | 1 |
| **Birth order** | | |
| 1st born | 58.3 | 7 |
| 2nd born | 33.3 | 4 |
| 3rd born | 8.3 | 1 |

cultural context, but the lack of financial resources poses a significant barrier to accessing proper medical care.

Emotional coping and financial strain were common challenges observed in the caregivers of children with sickle cell disease in Uganda, similarly to a study that was conducted in 2016 among the caregivers of SCD due to the endemicity of the disease in the country [1].

However, over time, many caregivers demonstrated a degree of acceptance and resilience in managing their emotions, resonating with existing literature on caregivers of children with chronic illnesses. This finding suggests that improving caregiver well-being involves providing psychosocial support to foster resilience and adaptable coping mechanisms to handle the stress caused by unplanned emergencies, frequent hospital visits, and lifestyle interruptions [15].

A study conducted in Bahrain, which used mixed methods, examined the social, emotional, and financial effects of caring for an adolescent with SCD on their caregivers. The findings revealed that lack of parking lots and traffic jams, as well as dissatisfaction with hospital facilities and a lack of adequate healthcare services, were the most common challenges faced by caregivers. More than half of the caregivers who preferred to seek care from smaller healthcare facilities were discouraged by these challenges, especially the extended wait in the emergency room [16]. This may differ from the current study, as Bahrain and Uganda have different economic statuses and healthcare approaches [17].

The participants' reliance on home-based remedies and traditional medicines highlights the importance of a holistic approach to care, which was also observed among Kenyan caregivers in a study facing several challenges in everyday tasks and physical, social, cognitive, and emotional well-being. Financial hardship was prevalent as a result of the carers' time spent caring for their children, leading to moderate to significant financial losses [18]. This can be attributed to the endemicity of SCD within the region in which both countries are located.

This study similarly highlights how some participants were actively stigmatized by both neighbors and the child's company, contributing to emotional distress and depression. In rural Kenya, a study on the gendered experiences of stigma in sickle cell disorder families found that lay practices of surveillance within affected families were associated with low initial recognition of SCD and its cause, contributing to stigmatization that occurred independently of genetic knowledge. Mothers were frequently held responsible, including possibly for incorrect paternity. Due to the loss of their independent means of support and the few options available to them for managing this long-term illness, mothers were often those most negatively impacted by SCD. [19].

The experiences of caregivers in this study demonstrated the initial confusion and uncertainty surrounding the disease before a formal diagnosis. Similarly, a quantitative study among African Americans that assessed their knowledge of SCD reported that 21% of participants had no prior knowledge of the condition. These findings underscore the need for more education and awareness for at-risk groups through non-gender-specific education, awareness, and screening campaigns aimed at vulnerable populations [20].

Financial strain and lack of support emerged as pervasive challenges impacting the caregivers' emotional well-being. To improve the well-being of caregivers of children with sickle cell disease in Uganda, it is crucial to address these challenges and provide adequate support, both financial and psychosocial. Policymakers and healthcare providers should consider implementing support programs that offer financial assistance to caregivers and provide accessible healthcare services to reduce the burden of medical expenses.

## Coping strategies

Differences can be observed in the level of social support and awareness. While some caregivers experienced negative reactions from their communities, others received emotional

encouragement from their support networks. These differences may be influenced by cultural norms and healthcare infrastructure in Uganda. Similarly in another study Social support played a crucial role in helping caregivers cope, but negative reactions from certain individuals highlighted the need for increased awareness and destigmatization efforts in the community in this study, In addition to describing the networking and support that caregivers sought to engage in for the purpose of support and information, the Lived Experience of Caregivers of Children with Sickle Cell Disease study also highlighted numerous parenting challenges that caregivers face while caring for their sickle cell disease children [21].

Coping strategies were diverse, with caregivers resorting to both traditional herbal remedies and modern medical treatments to alleviate pain and manage the disease's symptoms.

Unique themes shed light on cultural influences and individual perspectives. The positive attitude towards local herbal remedies reflects the integration of traditional and modern approaches to healthcare in Uganda, a similar study in knowledge of SCD among caregivers at a tertiary facility in Northern Ghana found that caregivers employed both pharmaceutical and non-pharmacological home management techniques, and some caregivers combined the two to control discomfort and keep an eye on their children' health. Despite the fact that the majority had previously used traditional medicine, they favor conventional interventions because they believe they are more successful [22]. Whereas for Uganda, the traditional remedies were used as an adjunct to modern medicine in cases where most of the participants were lacking enough funds and transport to the health facility according to this study.

Coping strategies were not limited to home-based remedies but extended to managing the child's condition at school as well. Caregivers demonstrated resourcefulness in ensuring their child's health needs were met both at home and in the educational setting. Similarly, as an addition to the qualitative study done in 2017 among the caregivers, that had identified a few coping mechanisms and on which self-education was among, this was perhaps due the time context of the study at the time of its conductance.

However, financial constraints and lack of adequate support emerged as significant stressors for the caregivers. Many participants faced financial hardships, impacting their ability to access proper medical care and support for their child's condition. This finding aligns with other studies conducted in similar contexts, where financial strain was reported as a common challenge for caregivers of children with sickle cell disease [1,22], this still shows a lot of economic hardships that are still prominent among the caregivers.

## Coping outcomes

The coping outcomes were multi-faceted. While caregivers showed resilience in their efforts to manage the disease, they also faced considerable emotional distress and sleep disruptions. The financial burden remained a significant source of stress, impacting their overall well-being. Also, compared to caregivers of children with low to mild depressive symptoms, caregivers of children with clinically elevated depressed symptoms displayed increased helplessness, according to one study conducted. In young people with sickle cell illness, parent suffering catastrophizing predicts a child's depressed symptoms [15].

Similarly, despite the challenges, caregivers showed resilience, drawing strength from their faith and hope for a better future for their children. Similarly, Caretakers indicated high levels of situational/demographic life stress and children's distractibility/hyperactivity in the home setting, but they also endorsed high levels of parental competence and acceptance of their children in another study that was conducted [23].

Emotional coping and encouragement were critical components of their coping efforts.

a similar Phenomenological Study, showed that Emotional coping strategies of Four out of six participants were somewhat hesitant to discuss the effects of caring for a child with SCD on their emotional/psychological health among the African Americans [24], whereas for this study, participants weren't hesitant to express their emotions, due to perhaps their willingness towards support for the common voices and empowerment.

The socioeconomic status (SES) and home surroundings of a cohort of carers of infants and toddlers with sickle cell disease (SCD) were also assessed in the study, along with stress and mental health symptoms. Parents of infants and toddlers (1–34 months) with SCD report significant levels of life stress but also high levels of self-awareness and parental ability. Despite the possibility of high levels of life stress in this population, caregivers of infants and toddlers with SCD were not found to exhibit any signs of psychological distress [23], in contrast to the findings of this study. These symptoms ranged from anxiety to difficulty falling asleep at night, most likely as a result of the prolonged period of time spent caring for the children and an older group for the children with SCD in this study.

The emphasis on parental love and support underscores the critical role of emotional bonding in care giving

## Strengths and limitations of the study

This study presents an exploration of the concealed narratives of caregivers of children living with sickle cell disease, delving into their experiences, coping strategies, and coping outcomes. The research was conducted in the Luganda language, enabling participants to express their feelings and perspectives more comprehensively. Pretesting of the interview questions was performed, and the study adhered to the COREQ guidelines [10].

Including participants aged 27 to 52 allowed for the examination of hidden stories, coping strategies, and coping outcomes across different generations. It is important to note that this study focused solely on SCD caregivers who received care at a single tertiary hospital in Uganda. While the findings may not be directly transferable to all caregiving contexts, the insights gained provide valuable perspectives on the experiences of caregivers in similar settings. Nevertheless, this hospital is the largest tertiary care facility in Uganda, providing services to the entire Ugandan population. Importantly, this study is the first of its kind in Uganda to investigate hidden stories, coping strategies, and coping outcomes.

Additionally, many participants experienced emotional breakdowns, which hindered their ability to fully share their untold stories.

## Conclusion

This study highlights the hidden stories shared by caregivers of children with Sickle Cell Disease (SCD) at the Mulago sickle cell clinic in Kampala, Uganda. These stories encompass financial burdens, long distances to healthcare facilities, and the knowledge and attitudes of family caregivers. Despite Mulago Hospital's role as a primary healthcare facility for SCD treatment, resource shortages and delayed diagnostic and treatment services further burden caregivers. Schools and religious centers offer support, but social stigma, neighborhood awareness gaps, diagnostic confusion, and the high financial strain contribute to a significant caregiver burden. To address these challenges, healthcare professionals and communities must recognize the caregivers' needs and include them in SCD care processes. Revising treatment guidelines and aligning them with the Sustainable Development Goal 3 for 2030 is essential for effective awareness campaigns and sustained behavioral change, ultimately reducing SCD-related mortality rates. This study also underscores the importance of advocating for

improved healthcare policies related to SCD. Its insights can inform initiatives that enhance access to services, alleviate caregiver burdens, and reduce healthcare expenses. The establishment of formal support systems, such as organizations or social communities, is critical to providing essential social, emotional, and financial assistance to families caring for adolescents with SCD.

## Supporting information

**S1 Table. Participants demographic characteristics.**
(PDF)

**S1 File. Alindas concept map.**
(DOCX)

## Acknowledgments

Special thanks go to Madam Lydia Kabiri for her invaluable guidance during the data collection and analysis process. I also extend my heartfelt gratitude to Dr. Ssebagala Hadad, the dedicated research assistant, and all the study participants for their valuable contributions to this research. I also wish to express my appreciation to the entire Owomugisha Vanah family for their unwavering support.

I would like to acknowledge the staff of Mulago National Referral Hospital for granting us permission to conduct our research within their hospital premises. Your cooperation has been instrumental in the success of this study.

## Author contributions

**Conceptualization:** Isaac Alinda.

**Data curation:** Isaac Alinda, Lydia Kabiri.

**Formal analysis:** Isaac Alinda, Lydia Kabiri.

**Funding acquisition:** Isaac Alinda, Lydia Kabiri.

**Investigation:** Isaac Alinda, Hadad Ssebagala.

**Methodology:** Isaac Alinda, Lydia Kabiri, Hadad Ssebagala.

**Project administration:** Isaac Alinda.

**Resources:** Isaac Alinda.

**Software:** Isaac Alinda.

**Supervision:** Lydia Kabiri.

**Validation:** Isaac Alinda.

**Writing – original draft:** Isaac Alinda.

**Writing – review & editing:** Isaac Alinda, Lydia Kabiri.

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
