## [Decision Letter · Decision Letter 0]

12 Mar 2024

PONE-D-23-37777Hidden stories of caregivers with children living with Sickle Cell Disease: experiences, coping strategies and outcomesPLOS ONE

Dear Dr. ALINDA,

Thank you for submitting your manuscript to PLOS ONE. After careful consideration, we feel that it has merit but does not fully meet PLOS ONE’s publication criteria as it currently stands. Therefore, we invite you to submit a revised version of the manuscript that addresses the points raised during the review process.

We look forward to receiving your revised manuscript.

Kind regards,

Deogratias Munube

Academic Editor

PLOS ONE

Journal Requirements:

2. Thank you for stating the following financial disclosure: "financial support granted by the Fogarty International center of the national institutes of health, US, department of state’s office of the U.S global AIDS co-coordinator and health diplomacy (S/GAC) and the president’s emergency plan for AIDS relief (PEPFAR), HEPI SHSSU (Health professions education and training for strengthening the health system and services in Uganda) under grant No. 1R25TW011213" 

3. Please amend your authorship list in your manuscript file to include author Hadadi Ssebagala.

4. We note you have included a table to which you do not refer in the text of your manuscript. Please ensure that you refer to Table 1 in your text; if accepted, production will need this reference to link the reader to the Table.

Additional Editor Comments:

Dear Author,

A decision of a major revision has been made of your manuscript.

Reviewers' comments:

Reviewer's Responses to Questions

**Comments to the Author**

1. Is the manuscript technically sound, and do the data support the conclusions?

Reviewer #1: Partly

2. Has the statistical analysis been performed appropriately and rigorously? 

Reviewer #1: N/A

3. Have the authors made all data underlying the findings in their manuscript fully available?

Reviewer #1: Yes

4. Is the manuscript presented in an intelligible fashion and written in standard English?

Reviewer #1: Yes

5. Review Comments to the Author

Reviewer #1: While I believe the manuscript is technically sound, some of the methods could benefit from more informed rationales.

Although the manuscript has been presented in an intelligible fashion and written in standard English, it will benefit from proofreading, which is highly recommended.

6. PLOS authors have the option to publish the peer review history of their article (what does this mean? ). If published, this will include your full peer review and any attached files.

**Do you want your identity to be public for this peer review?** For information about this choice, including consent withdrawal, please see our Privacy Policy .

Reviewer #1: No

---

## [Author Response · Author response to Decision Letter 1]

18 May 2024

Response to Reviewers:

We have carefully reviewed the comments and suggestions provided by the reviewers and journal staff. A detailed response to each comment, along with the revisions made in the manuscript to address these comments, is provided in the attached 'Response to Reviewers' document.

---

## [Decision Letter · Decision Letter 1]

29 Jul 2024

PONE-D-23-37777R1Hidden stories of caregivers with children living with Sickle Cell Disease in Uganda: experiences, coping strategies and outcomesPLOS ONE

Dear Dr. ALINDA,

Thank you for submitting your manuscript to PLOS ONE. After careful consideration, we feel that it has merit but does not fully meet PLOS ONE’s publication criteria as it currently stands. Therefore, we invite you to submit a revised version of the manuscript that addresses the points raised during the review process.

We look forward to receiving your revised manuscript.

Kind regards,

Aloysius Gonzaga Mubuuke

Academic Editor

PLOS ONE

Journal Requirements:

Additional Editor Comments:

None

Reviewers' comments:

Reviewer's Responses to Questions

**Comments to the Author**

1. If the authors have adequately addressed your comments raised in a previous round of review and you feel that this manuscript is now acceptable for publication, you may indicate that here to bypass the “Comments to the Author” section, enter your conflict of interest statement in the “Confidential to Editor” section, and submit your "Accept" recommendation.

Reviewer #1: (No Response)

Reviewer #2: (No Response)

2. Is the manuscript technically sound, and do the data support the conclusions?

Reviewer #1: Yes

Reviewer #2: Yes

3. Has the statistical analysis been performed appropriately and rigorously? 

Reviewer #1: Yes

Reviewer #2: N/A

4. Have the authors made all data underlying the findings in their manuscript fully available?

Reviewer #1: Yes

Reviewer #2: Yes

5. Is the manuscript presented in an intelligible fashion and written in standard English?

Reviewer #1: Yes

Reviewer #2: No

6. Review Comments to the Author

Reviewer #1: I am pleased that the authors have made some changes and/or provided further explanation to some sections of the manuscript. These have made the manuscript more insightful. There are a few more issues that need further attention, though.

1. While the authors have provided some background information on Uganda, they should give some specific details. For instance, how much does the treatment cost? What is the average income of families in Uganda?

Providing this information would aid readers in appreciating the financial challenges families face.

2. Member checking is not a required step for reflexive thematic analysis, and certainly, it is not done to ensure data saturation. Please, revise that statement in the methods section.

3. What is the rationale for summarizing the table showing the characteristics of the participants. There was no comment against the previous version. A reversion to the earlier format is recommended.

4. The authors maintain that reviewing extant literature is helpful to the interpretation of their data. I would be happy to have some citations that support this exercise.

5. I do not see ‘gender imbalance’ as a limitation – many of the caregivers are generally females so what you have is a true reflection of the population. In other words, you don’t have to state fewer male participation as a limitation.

6. On page 29, the main author has been described as “DM”. What does that mean?

Reviewer #2: Thanks to authors, they have attended to most of comments previously raised.

However, I feel the responses to the following could be improved.

Introduction

3. Taking care of a child with sickle cell disease can be expensive, and many families in Uganda may find it difficult to pay for the required therapies and drugs.” It will be more appropriate to provide contextual information on SCD and its management in Uganda in the ‘study setting’ section. The authors need to describe for instance what costs are incurred given that Mulago national referral hospital is a public facility.

Methods

7. How was the accuracy of data translation from Luganda to English ensured? The authors response “Consultation with Participants: Where necessary, the translated transcripts were reviewed with participants to ensure that the translated content accurately represented their original statements and meanings” is not clear to me. How this was practical given that participants were interviewed at the hospital while they had come to attend the clinic. Provide more details.

Results

1. I do not see the relevance of (n=12), (n=2) in the manuscript. Delete this style of reporting. Although, the authors had accepted the suggestion and deleted some, it was not implemented through the whole section. Please delete this style of reporting throughout the manuscript

3. I do not understand the sub-theme ‘coping strategies at school’. Such a phrase will be more suitable for the children’s experiences, rather than the experiences of caregivers. The authors’ responses are not conceiving. The issues covered here must directly be related to the study topic. I had for instance expected to find out how caregivers ensure that their children continue to take their treatment while at school, how do they ensure that medications are kept safely, get special permission to avoid stressful conditions, how they ensure that their children are able to attend clinic appointment dates during school term, any special diet arrangements at school etc Which is in line with quote from participant 4. The school fees challenges are cross cutting to most parents and is not specific to SCD caregivers!

Discussion

2. ‘Hidden’ or ‘concealed’ are not appropriate terms to use to describe the experiences of caregivers – consider using ‘untold’. The authors partially accepted the feedback. I re-echo the same. The authors should use consider using ‘untold’ instead of ‘hidden’ throughout the manuscript including the title.

Authors’ contributions

Reading authors’ contributions and acknowledgement section together leaves out confused. Both LK and HS are acknowledged for their input in this manuscript. HS was a research assistant and LK guided during the data collection and analysis process. How do they then turn around to be authors? Who si the hidden main author DM? Surprisingly, the contribution of IA is not shown.

Proof reading

The work will benefit from major proofreading as the errors are quite too many.

There are still many errors especially in the participants’ quotes. Actually, most of them are difficult to understand. E.g

".. I want him to go back [to school], because he is too worried, he says that the whole year, and yet in the future I will have like 2 years off…"

Participant 4, “…. things that helped me a lot after they had told me and I got to know that he is having sickle cell disease, the first thing I got to do was, I asked the Almighty to forgive me, since I decided it for myself, that I will never produce again, and by then I was pregnant preparing for the second baby, and then I said to myself that I would not produce a 2nd child because for the 1st child made me spend most of the times in the hospital while bed ridden

The participant was pregnant and is being quoted saying she will never ‘produce’ again. What happened or what plan did she have for the pregnancy?

This long sentence is very hard to comprehend: “Similarly in another study Social support played a crucial role in helping caregivers cope, but negative reactions from certain individuals highlighted the need for increased awareness and destigmatization efforts in the community in this study, In addition to describing the networking and support that caregivers sought to engage in for the purpose of support and information, the Lived Experience of Caregivers of Children with Sickle Cell Disease study also highlighted numerous parenting challenges that caregivers face while caring for their sickle cell disease children (20)”

Finally, under conclusion, authors refer to Mulago hospital’s role as a primary health care facility for SCD. However, in the previous sections, they had referred to Mulago hospital as a national referral hospital. The authors need to categorically state whether patients were attending Mulago SCD clinic as their primary care clinic or as a referral point.

Thank you

7. PLOS authors have the option to publish the peer review history of their article (what does this mean? ). If published, this will include your full peer review and any attached files.

**Do you want your identity to be public for this peer review?** For information about this choice, including consent withdrawal, please see our Privacy Policy .

Reviewer #1: No

Reviewer #2: **Yes: ** Vincent Mubangizi

---

## [Author Response · Author response to Decision Letter 2]

11 Sep 2024

Thank you for the opportunity to revise our manuscript based on the insightful comments provided by the reviewers and editors. We have carefully addressed each comment in the attached rebuttal letter, which provides a detailed explanation for the revisions made in response to each point raised.

In summary:

All comments from the reviewer have been fully addressed, including the addition of specific details regarding the cost of treatment, revisions to the methods section, updates to participant data presentation, and the correction of minor editorial points such as author initials.

Where applicable, citations were added to support interpretations, and changes were made in line with the journal's guidelines.

One additional reference was included in the manuscript, bringing the total to 27, and all references have been checked for accuracy.

We believe these revisions have improved the clarity and rigor of the manuscript, and we appreciate the feedback that facilitated these improvements. Please refer to the attached rebuttal letter for a point-by-point breakdown of all changes made.

---

## [Editor Report · Decision Letter 2]

7 Nov 2024

Hidden stories of caregivers with children living with Sickle Cell Disease in Uganda: experiences, coping strategies and outcomes

PONE-D-23-37777R2

Dear Dr. ALINDA,

We’re pleased to inform you that your manuscript has been judged scientifically suitable for publication and will be formally accepted for publication once it meets all outstanding technical requirements.

Kind regards,

Aloysius Gonzaga Mubuuke

Academic Editor

PLOS ONE
---

## [Editor Report · Acceptance letter]

PONE-D-23-37777R2

PLOS ONE

Dear Dr. ALINDA,

I'm pleased to inform you that your manuscript has been deemed suitable for publication in PLOS ONE. Congratulations! Your manuscript is now being handed over to our production team.

Kind regards,

on behalf of

Dr. Aloysius Gonzaga Mubuuke

Academic Editor

PLOS ONE